# The Formulation of Curcumin: 2-Hydroxypropyl-β-cyclodextrin Complex with Smart Hydrogel for Prolonged Release of Curcumin

**DOI:** 10.3390/pharmaceutics15020382

**Published:** 2023-01-22

**Authors:** Ljubiša Nikolić, Maja Urošević, Vesna Nikolić, Ivana Gajić, Ana Dinić, Vojkan Miljković, Srđan Rakić, Sanja Đokić, Jelena Kesić, Snežana Ilić-Stojanović, Goran Nikolić

**Affiliations:** 1Faculty of Technology, University of Niš, Bulevar Oslobodjenja 124, 16000 Leskovac, Serbia; 2Department of Physics, Faculty of Sciences, University of Novi Sad, Trg Dositeja Obradovica 4, 21000 Novi Sad, Serbia; 3Department of Chemistry, Biochemistry and Environmental Protection, Faculty of Sciences, University of Novi Sad, Trg Dositeja Obradovića 3, 21000 Novi Sad, Serbia

**Keywords:** curcumin, 2-hydroxypropyl-β-cyclodextrin, complex, N-isopropylmethacrylamide, N-isopropylacrylamide, smart hydrogel

## Abstract

Curcumin comes from the plant species *Curcuma longa* and shows numerous pharmacological activities. There are numerous curcumin formulations with gels or cyclodextrins in order to increase its solubility and bioavailability. This paper presents the formulation of complex of curcumin with 2-hydroxypropyl-β-cyclodextrin in a thermosensitive hydrogel, based on N-isopropylmethacrylamide and N-isopropylacrylamide with ethylene glycol dimethacrylate as a crosslinker. The product was characterized by chemical methods and also by FTIR, HPLC, DSC, SEM, XRD. The results show that synthesis was successfully done. With an increase in the quantity of crosslinker in the hydrogels, the starting release and the release rate of curcumin from the formulation of the complex with hydrogels decreases. The release rate of curcumin from the gel complex formulation is constant over time. It is possible to design a formulation that will release curcumin for more than 60 days. In order to determine the mechanism and kinetics of curcumin release, various mathematical models were applied by using the DDSolver package for Microsoft Excel application. The Korsmeyer-Peppas model best describes the release of curcumin from the gel formulation of the complex, while the values for the diffusion exponent (0.063–0.074) shows that mechanism of the release rate is based on diffusion.

## 1. Introduction

Curcumin ((1E,6E)-1,7-bis(4-hydroksy-3-methoxyphenyl)-1,6-heptadiene-3,5-dione), Figure 1, is a natural polyphenol isolated from the rhizome of the plant species *Curcuma longa* which shows numerous pharmacological activities by the modulation of physiological and biochemical processes. Research shows that curcumin possesses hypoglycemic [1], antimicrobial [2], hepatoprotective [3], anti-inflammatory [4], antioxidant [5], anticancer [6], antiviral [7] and many other effects. Various animal studies and clinical trials have shown that curcumin is safe to apply at high doses because it does not affect liver and kidney function. However, due to poor water solubility and low bioavailability, it is classified as a drug of group IV by the biopharmaceutical classification system, and its therapeutic application is limited [8,9,10,11]. In order to improve the physico-chemical properties, different systems for curcumin delivery were used: cyclodextrins for the formation of inclusion complexes [12,13,14,15], micelles [16], liposomes [17,18], nanoemulsions [19], hydrogels [20], polymers formed by cross-linking of cyclodextrin [21,22,23], polymer nanofibers [24,25,26], complexes with cyclodextrin incorporated into polymers [27,28,29] and other nanoparticles [2,30]. In the study of Purpura et al., the bioavailability of formulation of curcumin with β-cyclodextrin was examined. The results of the study show that the formulation of β-cyclodextrin with curcumin significantly improves the absorption of curcumin in healthy people [31]. In the experiment of Jafer et al., with the aim of improving the delivery of curcumin in the treatment of cancer cells, an inclusion complex of β-cyclodextrin with curcumin was prepared. The obtained results show that the complex of β-cyclodextrin with curcumin improved the delivery and antiproliferative effect to the MCF-7 breast cancer cells [14]. The inclusion complex of curcumin with β-cyclodextrin obtained by coprecipitation method increased the water solubility of curcumin from 0.00122 to 0.721 mg/cm^3^. The release of the inclusion complex from nanocomposite and conventional poly(N-isopropylacrylamide/sodium alginate) hydrogels crosslinked with N,N′methylenebis(acrylamide) (BIS), respectively, was tested under simulated gastrointestinal conditions. At pH = 1.2, hydrogels showed the lowest release and swelling ratio, but at pH = 6.8 the highest release to swelling ratios of curcumin were achieved [28]. A thermosensitive hydrogel was synthesized, poly(D,L-lactide-co-glycolide)-poly(ethylene-glycol)-poly(D,L-lactide-co-glycolide), as a carrier for the delivery of doxorubicin in combination with an inclusion complex of curcumin with β-cyclodextrin for the treatment of cancer cells. Combined therapy based on doxorubicin and inclusion complex of curcumin with β-cyclodextrin showed greater antitumor activity than monotherapy in vitro [29]. Zhang et al., synthesized in situ forming hydrogels based on polyvinyl pyrrolidone, encapsulated with a solid dispersion of curcumin for the healing of vaginal wounds and the treatment of vaginal bacterial infections. After local application for treatment of the vaginal infection caused by *Escherichia coli* and *Staphylococcus aureus*, high efficiency in therapeutic treatment has been confirmed, along with inflammation reduction and improved healing of vaginal wounds [32]. Shefa et al. have synthesized a biocompatible and biodegradable hydrogel system for the delivery of curcumin based on polyvinyl alcohol and oxidized cellulose nanofibers, in order to improve the wound healing process [33]. Thermosensitive β-glycerophosphate/chitosan hydrogels, with an encapsulated complex of curcumin:β-cyclodextrin for the treatment of skin wound infections, were synthesized in work of Zao et al. The ability of wound healing using the above mentioned hydrogel was tested on induced superficial wounds in rats. By analyzing the results, it was observed that the wounds treated with hydrogel containing the complex of curcumin:β-cyclodextrin showed a faster healing rate compared to wounds that were covered only with gauze [34]. A drug delivery system based on poly(N-isopropylacrylamide), p(NiPAm), hydrogel and a suitable solvent for improvement of solubility and local release of curcumin was also synthesized in the experimental work of Ayar et al. Curcumin was incorporated in p(NiPAm) hydrogel during swelling by using methanol or polyethylene glycol of low molecular weight (PEG200). The obtained results show that PEG200 increases curcumin solubility more than methanol, and shows a superior effect on the cumulative quantitative of curcumin released over 7 days (33.163  ±  0.319 mg/cm^3^) compared to methanol (8.765  ±  0.544 mg/cm^3^). P(NiPAm) hydrogel combined with PEG200 did not show any cytotoxicity, and can be used as an effective sustained release system for curcumin [35].

In recent years, highly advanced curcumin delivery systems have been developed such as nanoparticles, ultrasound microbubbles, exosomes, biopolymer nanoparticles [36], nanogels, nanosuspensions, nanoemulsions and dendrimer [37]. The goal for creating such formulations was to improve the stability and solubility of curcumin.

The polymer hydrogels based on N- isopropylmethacrylamide (NiPMAm) and/or N-isopropylacrylamide (NiPAm) are well known because of their good property to respond to temperature changes by changing the degree of swelling [38,39]. To create a three-dimensional network of these gels, ethylene glycol dimethacrylate (EGDM) is often used as a networker. If in their structure they contain a copolymerized organic acid (acrylic acid, methacrylic acid, etc.), the degree of swelling of these gels will also change with the change in the pH value of the environment [40]. That is why these gels are called smart hydrogels, because they react by changing the degree of swelling with the change of the conditional parameters in which they are found. Namely, if the temperature of these swollen hydrogels is increased, their degree of swelling decreases and they squeeze out liquid and vice versa, but if their pH value is lowered the degree of swelling increases and vice versa. These kinds of gels have been used as matrix systems for medicinal substances [41]. In Figure 2, chemical structures of N-isopropylmethacrylamide, N- isopropylacrylamide and ethylene glycol dimethacrylate are shown.

The topic of this paper is the development of a new matrix system based on a smart polymer carrier crosslinked with poly(N-isopropylmethacrylamide-co-N-isopropylacrylamide), with an incorporated inclusion complex of curcumin: 2-hydroxypropyl-β-cyclodextrin, with the aim of increasing the solubility of curcumin. With this system, targeted delivery and sustained release of curcumin can be performed. In Figure 3, part of a cyclic structure of a seven-membered ring molecule of 2-hydroxypropyl-β-cyclodextrin is shown.

## 2. Materials and Methods

### 2.1. Reagents

N-isopropylmethacrylamide (NiPMAm) 97%, N-isopropylacrylamide (NiPAm) 99%, 2,2′-azobis(2-methylpropionitrile) (AIBN) 98% (Acros Organics, New Jersey, NJ, USA); ethylene glycol dimethacrylate (EGDM) 97% (Fluka Chemical Corp, Buchs, Switzerland); curcumin (CU) 97%, 2-hydroxypropyl-β-cyclodextrin (2-HP-β-CD) 97% (Tokyo Chemical Industry Co., Ltd., Tokyo, Japan); Tween 20 (Alfa Aesar, ThermoFisher, Kandel, Germany); potassium bromide (KBr) 99%, ethanol (Et) 99.5%, methanol for HPLC (Me) ≥99.9% (Merck KGaA, Darmstadt, Germany); Hanks’ buffered solution pH 7.4 GmbH (PAA Laboratories, Pasching, Austria). All reagents were used with no further purification.

### 2.2. Synthesis of Hydrogels

Hydrogels samples were synthesized according to the procedure described previously [39]. In short, hydrogels poly(N-isopropylmethacrylamide-*co*-N-isopropylacrylamide), p(NiPMAm/NiPAm), in the molar ratio 10/90 were synthesized by radical polymerization of monomers NiPMAm and NiPAm, with EGDM as a crosslinker in 2 and 3 mol% relative to the total amount of monomer. Obtained gels were marked 10/90/2 and 10/90/3, respectively. Ethanol was used as a solvent and 30 mg of AIBN for initiation of the polymerization reaction. After dissolving the reactants, the homogenized reaction mixtures were injected into glass ampoules which were then heat sealed. The polymerization reaction was performed in the mode: 0.5 h at 70 °C, 2 h at 80 °C and 0.5 h at 85 °C. After polymerization and cooling of the samples at room temperature, copolymerized p(NiPMAm/NiPAm) hydrogels were obtained in the form of long cylinders and cut into discs of thickness 5 mm. Hydrogels prepared in this way were extracted with methanol during 168 h in order to remove unreacted reactants, then washed off with water for 24 h to remove methanol and after that, dried at 40 °C to constant weight.

### 2.3. Lyophilization of Gels

Lyophilization p(NiPMAm/NiPAm) of hydrogels swollen up to the equilibrium was performed on the apparatus LH Leybold Heraeus, Lyovac GT2 (Labexchange, Frekendorf, Switzerland). The synthesized hydrogels are firstly frozen at a temperature of −40 °C during 24 h. In first subphase of drying, the volume of solution was reduced by sublimation at −30 °C and pressure of 5 Pa during 12 h. In the second subphase of drying, i.e., isothermal desorption, the hydrogels were heated up to 20 °C during 6 h at the pressure of 5 Pa, with removal of steam. Lyophilized samples of hydrogels were packed under vacuum condition and stored in a refrigerator at 5 °C.

### 2.4. Obtaining of the Complex

Curcumin (368.38 mg) was dissolved in 200 cm^3^ of absolute ethanol and added to the solution of 2-hydroxypropyl-β-cyclodextrin obtained by dissolution of 1541.54 mg 2-hydroxypropyl-β-cyclodextrin in 100 cm^3^ of distilled water. The mixture obtained like this was equilibrated by mixing on a magnetic stirrer at room temperature during 96 h, protected from light. The resulting solution was concentrated on a vacuum evaporator at 40 °C to the minimum volume, and then dried in a desiccator over a dehydrating agent at room temperature to constant mass. The molar ratio of curcumin and 2-hydroxypropyl-β-cyclodextrin in inclusion complex was 1:1.

### 2.5. Phase Solubility

Phase solubility study was performed according to method described by Higuchi & Connors [42]. Surplus of curcumin (each 10 mg) was added in 2.5 cm^3^ water solution of 2-hydroxypropyl-β-cyclodextrin of concentration 0–10 mmol/dm^3^. The samples were stirred at room temperature during 24 h, and then filtered through a membrane filter with a pore diameter 0.45 μm (Econofilters, Agilent Technologies, Waldborn, Germany). The quantity of dissolved curcumin was determined by application of UV/V method on the basis of constructed calibration curves of absorbance dependence on concentration. Measurements were performed by spectrophotometer Cary-100 Conc (Varian PTY LTD, Springvale, Australia) at wavelength 429 nm, in quartz cuvettes (1 × 1 × 4.5 cm) at room temperature. Distilled water was used as a blank solution. The presence of 2-hydroxypropyl-β-cyclodextrin does not affect the absorbance of curcumin at 429 nm, because its absorption at that wavelength is equal to 0. The constant of stability (K_1:1_) of inclusion complex was calculated based on the phase solubility diagram according to Equation (1):(1)$$K_{1:1}=\frac{slope}{S_{0}\left(1-slope\right)},$$
where *S*_0_ is solubility of curcumin at 25 °C in the absence of cyclodextrin (intercept) and slope represents the value from the phase solubility graph.

### 2.6. Incorporation of Complex of Curcumin: 2-Hydroxypropyl-β-cyclodextrin into p(NiPMAm/NiPAm) Gels

Matrix systems with a curcumin: 2-hydroxypropyl-β-cyclodextrin complex and p(NiPMAm/NiPAm) hydrogels were obtained by swelling the hydrogels to equilibrium in ethanol in which the inclusion complex of curcumin: 2-hydroxypropyl-β-cyclodextrin in concentration of 1.23 mg/cm^3^ was dissolved, at a temperature of 25 °C. Weighed samples of xerogels p(NiPMAm/NiPAm), 50 mg, were covered with a solution of curcumin inclusion complex (5 cm^3^) and left to swell at room temperature, protected from light. After reaching equilibrium, swollen p(NiPMAm/NiPAm) the hydrogels were separated from the remaining solution by decantation, washed with distilled water in order to remove the unincorporated amount of the inclusion complex of curcumin: 2-hydroxypropyl-β-cyclodextrin, and after that extra water was removed from the surface of the matrix gels. The quantity of incorporated curcumin in p(NiPMAm/NiPAm) was determined based on the difference in the quantities of curcumin in the initial solution of the inclusion complex curcumin: 2-hydroxypropyl-β-cyclodextrin, and the supernatant after equilibrium was reached by using the HPLC method. The efficiency of incorporation of curcumin in hydrogel, η, was calculated according to Equation (2):(2)$$\eta \left(\%\right)=\frac{L_{g}}{L_{u}}\;\cdot \;100,$$
where *L_g_* is mass of curcumin incorporated in p(NiPMAm/NiPAm) hydrogel, mg/g_xerogel_, and *L_u_* initial mass of curcumin entered by the solution of the curcumin: 2-hydroxypropyl-β-cyclodextrin inclusion complex for swelling and incorporating into xerogel, mg/g_xerogel_.

The schematic view of obtaining the formulation of curcumin: 2-hydroxypropyl-β-cyclodextrin complex with p(NiPMAm/NiPAm) hydrogel is shown in Figure 4.

### 2.7. The Release of Curcumin from Matrix System

In vitro study of the release of curcumin from swollen p(NiPMAm/NiPAm) hydrogels with incorporated inclusion complex of curcumin: 2-hydroxypropyl-β-cyclodextrin was carried out in a medium that simulates physiological conditions. Each sample was covered with 10 cm^3^ of solution (9 cm^3^ Hank’s BSS buffer with pH value 7.4 and 1 cm^3^ of Tween 20 solution concentration of 1.52 mg/cm^3^). The samples were thermostated in a water bath at 37 °C with stirring on a magnetic stirrer (Hanna Instruments, Magnetic stirrer HI 190M) during 48 h. The amount of curcumin released was monitored by sampling 200 µL of solution over time, which were then filled up with methanol up to 1 cm^3^, filtered on a cellulose membrane filter with a pore diameter 0.45 µm and analyzed by HPLC method. The kinetics of the curcumin release from the matrix system was evaluated by different mathematical models (Higuchi, Korsmeyer–Peppas, Baker–Lonsdale) with DDSolver package for Microsoft Excel applications.

### 2.8. Determination of the Concentration of Some Compounds by Using High Pressure Liquid Chromatography (HPLC)

The content of residual reactant (monomers and crosslinkers) in samples of synthesized p(NiPMAm/NiPAm) hydrogels was calculated by HPLC method. The analysis was performed by using the apparatus HPLC Agilent 1100 Series (Waldborn, D) equipped with diode-array detector, DAD 1200 Series. Conditions for chromatography performance: column Zorbax Eclipse XDB-C18 (4.6 × 250 mm, 5 μm) (Agilent Technologies, Inc., Santa Clara, CA, USA); temperature 25 °C; injected volume of samples 10 μL; detection wavelength 210 nm; mobile phase consists of methanol/redistilled water 70/30, *v*/*v*; mobile phase flow was 0.5 cm^3^/min. The results obtained were processed by software Agilent Chemstation. On the basis of constructed calibration curves for the linear dependence, the equations were obtained for determining the content of NiPMAm, NiPAm and EGDM in methanol extracts obtained by the processing of synthesized p(NiPMAm/NiPAm) hydrogels.

The dependence of the peak area on the concentration of NiPMAm is linear in range of 0.005–0.250 mg/cm^3^. For the straight part of the calibration curve of NiPMAm the Equation (3) applies with a linear correlation coefficient R^2^ = 0.995.
(3)$$c=\frac{A-572.14}{97110.1},$$

Dependence of peak area on concentration of NiPAm is linear in range of 0.005–0.250 mg/cm^3^ and in the straight part of the calibration curve the Equation (4) applies, R^2^ = 0.997.
(4)$$c=\frac{A-594.72}{137938.4},$$

Dependence of peak area on concentration of EGDM is linear in range 0.005–0.250 mg/cm^3^. For the straight part of the calibration curve of EGDM the Equation (5) applies, R^2^ = 0.998.
(5)$$c=\frac{A-911.18}{171,931},$$

In Equations (2)–(4), A is pick area (mAU·s), and c is concentration of reactants (mg/cm^3^) NiPMAm, NiPAm either EGDM.

Determination of the quantity of curcumin that was not incorporated into the hydrogels, as well as monitoring of curcumin release from the hydrogels, was performed by using liquid chromatography HPLC method at these conditions: column Zorbax Eclipse XDB-CN 250 × 4.6 mm, 5 μm (Agilent Technologies, Inc., Santa Clara, CA, USA); eluent was methanol: mobile phase flow was 1 cm^3^/min; volume of injected samples 20 μL; column temperature 40 °C; detection wavelength 425 nm. For the straight line part of the curcumin calibration curve in range 0.53–106 µg/cm^3^, Equation (6) applies with a linear correlation coefficient R^2^ = 0.999.
(6)$$c=\frac{A+158.26}{184.15},$$
where A is pickk area (mAU·s), and c is concentration of curcumin (µg/cm^3^).

### 2.9. Swelling of Hydrogels

Swelling of synthesized p(NiPMAm/NiPAm) xerogels was monitored gravimetrically. A known quantity of p(NiPMAm/NiPAm) xerogels was immersed in a water solution of a certain pH value and temperature, and then the mass of the sample was measured at certain time intervals until equilibrium was reached, i.e., until constant mass of hydrogels was reached. Aqueous mediums for swelling were prepared by adjusting the pH value by addition of 0.1 M solution of sodium hydroxide (Centrohem, Beograd, Serbia) or 0.1 M solution of hydrochloric acid (Zorka, Šabac, Serbia) while observing the value on pH meter (HI9318-HI9219, Hanna, P). The thermosensitivity of hydrogels was tested in the temperature range of 25 to 80 °C in a water bath. The degree of swelling, α, was calculated according to Equation (7).
(7)$$\alpha =\frac{m-m_{0}}{m_{0}},$$
where *m_0_*—is mass of dry gel, *m*—mass of swollen gel in a point of time t.

To analyze the type of solvent diffusion process inside hydrogels, Equation (8) applies, which stands for condition (*Mt*/*Me* ≤ 0.6) [43,44]:(8)$$F=\frac{M_{t}}{M_{e}}=k\;\cdot \;t^{n},$$
where *F*—is the fractional sorption, *M_t_*—mass of the absorbed solvent in a point of time, *M_e_*—mass of the absorbed solvent in equilibrium, *k*—a constant that is characteristic for a certain type of polymer network (min^1/*n*^) and *n*—diffusion exponent. By logarithmizing Equation (8) comes a linear Equation (9) that can be applied to calculate the constant *k* and an exponent *n*.
(9)$$\ln\frac{M_{t}}{M_{e}}=\ln k+n\;\cdot \;\ln t,$$

The value of diffusion exponent *n* determines the mechanism of fluid diffusion. For value *n* = 0.5 the fluid diffusion mechanism corresponds to Fick’s law of diffusion (case I), where the rate of solvent transport into the gel is lower than the rate of relaxation of polymer chains. When the value for diffusion exponent is lower than 0.5, the penetration of solvent is much slower than the polymer chains relaxation. The solvent transportation mechanism is a part of Fick’s diffusion and is called “less to Fick’s” diffusion. Anomalous diffusion mechanism (non-Fick’s diffusion) occurs with 0.5 < *n* < 1, and then the hydrogels swelling is under control by both solvent diffusion in matrix and polymer chains relaxation. The diffusion process is a lot faster when the diffusion exponent has value 1 in comparison to the polymer system chains relaxation (case II), while when *n* > 1 the polymer chains relaxation is under control of gels swelling (case III, Super case II). The diffusion coefficient of solvent molecules in hydrogel (*D*) is most often determined by applying the Equation (10) which takes into account only the initial stage of swelling (60% of swelling) during which the thickness of the polymer remains approximately constant [45,46]:(10)$$\frac{M_{t}}{M_{e}}=4\;\cdot \;{\left(\frac{Dt}{\pi l^{2}}\right)}^{1/2},$$
where *D* stands for diffusion coefficient (cm^2^/min), and *l* for the thickness of the dry hydrogel, xerogel (cm). By logarithmization of Equation (10), linear dependence comes between ln(*M_t_/M_e_*) and ln*t* (Equation (11)) from the section of which the diffusion coefficient *D* is calculated.
(11)$$\ln\frac{M_{t}}{M_{e}}=\left(\frac{4D^{1/2}}{\pi^{1/2}l}\right)+\frac{1}{2}\ln t,$$

### 2.10. Fourier Transform Infrared Spectroscopy (FTIR)

The FTIR spectra of monomers, synthesized p(NiPMAm/NiPAm) xerogels, curcumin, 2-hydroxypropyl-β-cyclodextrin, inclusion complex of curcumin: 2-hydroxypropyl-β-cyclodextrin, xerogels with incorporated complex of curcumin: 2-hydroxypropyl-β-cyclodextrin, were recorded by the thin transparent tablets technique with potassium bromide of spectroscopic purity, by vacuuming and pressing under a pressure of approximately 200 MPa. The preparation of tablets was measured by 150 mg of potassium bromide and 0.7 mg each sample which were pulverized on an amalgamator (WIG-L-Bug, Dentsply RINN, a Division of Dentsply International Inc., York, PA, USA). Crosslinker EGDM was recorded in the form of a thin film between two plates of zinc selenide (ZnSe). The recording was performed on FTIR spectrophotometer BOMEM MB-100 (Hartmann & Braun, Baptiste, Quebec, QC, Canada) in the wavenumber range of 4000 to 400 cm^−1^. Spectra were processed by application Win-Bomem Easy software.

### 2.11. Scanning Electron Microscopy (SEM)

The morphology of curcumin, complex of curcumin: 2-hydroxypropyl-β-cyclodextrin, lyophilized p(NiPMAm/NiPAm) hydrogels and p(NiPMAm/NiPAm) hydrogels with incorporated complex of curcumin: 2-hydroxypropyl-β-cyclodextrin was examined by scanning electron microscopy. Before analysis, the pulverized sample was coated with gold/palladium alloy (15/85) by using a sprayer JEOL Fine Coat JFC 1100E Ion Sputter (JEOL Ltd., Tokyo, Japan) and recorded on apparatus JEOL Scaning Electron Microscope JSM-5300 (JEOL Ltd., Tokyo, Japan).

### 2.12. Differential Scanning Calorimetry (DSC)

For testing the thermal properties of curcumin, 2-hydroxypropyl-β-cyclodextrin, inclusion complex of curcumin: 2-hydroxypropyl-β-cyclodextrin, empty p(NiPMAm/NiPAm) hydrogels and hydrogels with incorporated complex of curcumin: 2-hydroxypropyl-β-cyclodextrin differential scanning calorimetry method was applied. The sample (about 3 mg) was placed in a vessel and heated in one cycle from room temperature up to 250 °C heating dynamics 10 °C/min in a nitrogen atmosphere. This testing was performed by using apparatus differential scanning calorimeter TA Instruments Q20 (TA Instruments, New Castle, DE, USA).

### 2.13. X-ray Diffraction (XRD)

XRD spectra of curcumin, 2-hydroxypropyl-β-cyclodextrin, inclusion complex of curcumin: 2-hydroxypropyl-β-cyclodextrin, empty p(NiPMAm/NiPAm) hydrogels and hydrogels with incorporated complex of curcumin: 2-hydroxypropyl-β-cyclodextrin were recorded at these conditions: samples were marked by monochrome CuKα radiation and analyzed at an angle 2θ between 5 and 75° with a sequence of 0.05° and recording time τ = 5 s. During recording, voltage and current used were 40 kV and 20 mA, respectively. All tested samples were recorded by a powder diffractometer Rigaku MiniFlex 600 (Rigaku, Tokyo, Japan).

### 2.14. Nuclear Magnetic Resonance (^1^H-NMR)

^1^H-NMR spectra of 2-hydroxypropyl-β-cyclodextrin and of curcumin: 2-hydroxypropyl-β-cyclodextrin inclusion complex were recorded on Bruker Avance III NMR, 400 MHz (BRUKER AXS GmbH, Karlsruche, Deutschland) apparatus in a glass cuvette with a diameter of 5 mm at room temperature by the pulse method, with multiple repetition of pulses. The samples were dissolved in deuterated water (D_2_O) and the solutions were treated in an ultrasonic bath for 25 min before recording.

## 3. Results

### 3.1. Phase Solubility

Phase solubility analysis was performed in order to determine and compare the solvation and complexation power of 2-hydroxypropyl-β-cyclodextrin to curcumin. From Figure 5 it can be seen that phase solubility diagram is of “A_L_” type [42]. This indicates that the molar ratio between host and guest molecule in the inclusion complex is 1:1, and that the solubility of curcumin is increasing linearly with the increase of 2-hydroxypropyl-β-cyclodextrin concentration. By linear fitting, using the data from Figure 5 the following is obtained: slope = 9.953 × 10^−4^, intercept = 8.571 × 10^−6^ mol/dm^3^. By using the data from Figure 5 for slope and intercept and Equation (1), the value of the stability constant of the curcumin and 2-hydroxypropyl-β-cyclodextrin complex was calculated: K_1:1_ = 116.23, and solubilization efficiency (ratio of curcumin solubility in water solution in presence of highest tested concentration of 2-hydroxypropyl-β-cyclodextrin, 10 mmol/dm^3^, and solubility of pure curcumin itself in water) was 1237.18.

### 3.2. The Swelling

Dependence of the degree of swelling on time of p(NiPMAm/NiPAm) of lyophilized hydrogels (10/90/2 i 10/90/3) in solution with pH = 7.4 on different temperatures (25 and 37 °C) is shown in Figure 6a,b, one after the other. From these experiments, the equilibrium values of the degree of swelling for the corresponding gels and temperatures was determined. The dependence of the degree of swelling on temperature is shown in Figure 7, and it can be seen that degree of swelling decreases when temperature increases. There is a phase transition—the lower critical temperature of solution (LCST), at a temperature of approximately 37 °C for this hydrogel composition—when hydrogel transforms from hydrophilic to hydrophobic form. At temperatures above the temperature of phase transition, the breakup of hydrogen bonds with water molecules occurs, the hydrophobic interactions become dominant and the polymer network contracts [47].

That is why this type of hydrogel has a higher degree of swelling at lower temperatures, and this property can be used to enter a larger amount of active substance into the hydrogel structure at lower temperatures. On the contrary, when the temperature increases, these hydrogels have a lower swelling degree. This means that they show a tendency to squeeze some of the liquid out of their structure. Therefore, the concept of this work was to conduct the experiment on the release of curcumin from the hydrogel complex formulation at a temperature of 37 °C, when equilibrium is established in the structure throughout the time of swelling.

In Table 1 the values are shown for kinetic parameters: the constants that are characteristic for certain types of polymer network, diffusion exponents and diffusion coefficients for swelling of p(NiPMAm/NiPAm) hydrogels at 25 and 37 °C with pH 7.4, obtained by the application of Equations (9) and (11).

The transport of solution into the polymeric matrix of p(NiPMAm/NiPAm) hydrogels at a temperature 25 °C and pH 7.4 presents the anomalous type of diffusion (non-Fick’s diffusion), where the value of the diffusion exponent has to be in the range of 0.5 < *n* < 1. For the tested samples it is in the range 0.60–0.82. At a temperature of 37 °C and pH = 7.4, the swelling process of p(NiPMAm/NiPAm) hydrogels 10/90/2 is determined by the diffusion of the aqueous solution (“less to Fick’s” diffusion, *n* = 0.49 ≈ 0.5), while for the hydrogel sample, 10/90/3 solvent diffusion into the matrix and the polymer chain relaxation controls the swelling because of the diffusion exponent value 0.63.

### 3.3. Residual Monomers Analysis

By using the HPLC method, the content of unreacted reactants in p(NiPMAm/NiPAm) hydrogels synthesis process was determined, using calibration curves for monomers (NiPMAm and NiPAm) and crosslinker (EGDM). At selected conditions for chromatographic analysis the retention time (R_t_) for monomer NiPMAm was 6.693 min, NiPAm 6.176 min and crosslinker EGDM 12.812 min. Residual quantities of monomers in copolymer samples of p(NiPMAm/NiPAm) in relation to their initial quantity in the reaction mixture are shown in Table 2.

Since the toxicity of the residual monomers is limited by their content in the copolymer, and that in the synthesized samples their content is acceptable (<0.3%), synthesized p(NiPMAm/NiPAm) hydrogels can be considered as safe for use as carriers for bioactive substances.

The procedure with variable temperature used for the synthesis of these hydrogels, which was developed through research in the previous period, enables such a low content of residual monomers. This is the minimum amount of residual monomers achieved by the synthesis. Hydrogels can even get rid of these low amounts of residual monomers by extraction with methanol and then washing with water. In this way, a hydrogel without residual monomers will be obtained, which is the most favorable for the production of pharmaceutical formulations.

### 3.4. FTIR Spectroscopy Analysis

Structural analyses of the starting monomers NiPMAm, NiPAm, crosslinker EGDM and synthesized 2 mol% EGDM p(NiPMAm/NiPAm) xerogel were carried out by using the FTIR method (Figure 8). In addition, the structure of the inclusion complex of curcumin: 2-hydroxypropyl-β-cyclodextrin, as well as the matrix system, p(NiPMAm/NiPAm) gel in which was incorporated curcumin: 2-hydroxypropyl-β-cyclodextrin inclusion complex were examined by using this method. In Figure 8, FTIR spectra of NiPMAm monomers, comonomers NiPAm, crosslinker EGDM and synthesized p(NiPMAm/NiPAm) xerogel containing 2 mol% of EGDM are shown.

In the FTIR spectrum of NiPMAm monomer (Figure 8a), the absorption band 3291 cm^−1^ is a result of valence N-H vibrations, ν(N-H). Asymmetric valence vibrations of C-H bond of the vinyl group, νas(=C-H), give an absorption band with a maximum at 3061 cm^−1^. The absorption bands at 2973 and 2878 cm^−1^ are from asymmetric and symmetric valence of the methyl group in the NiPMAm monomer structure, respectively. The proof for the presence of the amide group in the monomer structure are absorption bands with maximums at 1653 cm^−1^ (Amide band I) and 1539 cm^−1^ (Amide band II). It can be assumed that amide band I is from the valence vibrations of the keto group, while Amide band II arises by coupling of N-H deformation vibrations and valence C-N vibrations. The valence vibrations of C=C bond in FTIR spectrum of NiPMAm monomer (Figure 8a) give an absorption band with a maximum at 1606 cm^−1^. The absorption band of asymmetric deformative C-H vibration in the plane of CH_3_-C group in the monomer spectrum is present at 1459 cm^−1^. The valence vibrations of C-H bond of isopropyl group give an absorption band with a maximum at 1363 cm^−1^. The presence of the isopropyl group in the monomer structure is also confirmed by the presence of an absorption band at 1157 and 1011 cm^−1^.

In FTIR spectrum of NiPAm comonomers in the wavelength range of 3000 cm^−1^ (Figure 8b), two absorption bands with different intensity can be observed. The high intensity absorption band with a maximum at 3295 cm^−1^ is attributed to valence vibrations of the secondary amino group, ν(N-H) which is in agreement with the research of other authors [48], while the absorption band at 3073 cm^−1^, is a result of asymmetric vibrations of vinyl group, νas(=C-H). In FTIR spectrum of NiPAm comonomer (Figure 8b), absorption bands with a maximum at 2971 cm^−1^ and 2876 cm^−1^ come from asymmetric and symmetric valence vibrations of C-H bond from the methyl group, respectively. The absorption band of medium intensity with a maximum at 2934 cm^−1^ comes from asymmetric valence vibrations, ν_as_(C-H), of C-H bonds in isopropyl group of NiPAm. The Amide bands I, II and III with a maximum at 1659 cm^−1^, 1549 cm^−1^, 1310 cm^−1^, respectively, confirm the presence of an amide group in the molecule of NIPAM [49]. The valence vibrations of C=C bond in FTIR spectrum of NiPAm monomer (Figure 8b) provides an absorption band with a maximum at 1619 cm^−1^. The absorption band of medium intensity at 1371 cm^−1^, relates to deformation vibrations in the plane δ(C-H), of C-H bond in isopropyl group of NiPAm. The high intensity band with a maximum at 1167 cm^−1^ in the comonomer spectrum also confirms the presence of the isopropyl group in the structure of NiPAm. The appearance of absorption bands which come from deformation vibrations in the plane δ(=C-H), at 1412 cm^−1^ and deformation vibrations out of the plane γ(=C-H), at 990 and 917 cm^−1^ confirms the presence of the vinyl group in the structure of the comonomer [49].

In the FTIR spectrum of crosslinker EGDM (Figure 8c), there are absorption bands characteristic for ester and vinyl functional groups present in the molecule. Sharp absorption bands of high intensity with a maximum at 1723 cm^−1^ in FTIR spectrum of EGDM, comes from valence vibrations of the carbonyl group that is conjugated by a double bond, which is why the medium intensity band that corresponds to the valence vibrations of the C=C bond is noticeable at 1637 cm^−1^. The valence vibrations of C-O bond give an absorption band with a maximum of absorption at 1154 cm^−1^. In the FTIR spectrum of crosslinker EGDM, absorption bands with a maximum at 2894 cm^−1^ from ν_s_(CH_3_), at 2961 cm^−1^ from ν_as_(CH_3_), at 2930 cm^−1^ from ν_as_(CH_2_) and at 3106 cm^−1^ from the vinyl group ν_as_(=CH) are present, which is in accordance with the literature data [39].

By comparing the FTIR spectrum of p(NiPMAm/NiPAm) copolymer with 2 mol% of EGDM (Figure 8d) to the spectrum of NiPMAm monomer (Figure 8a) and NiPAm (Figure 8b), there is a clear difference that indicates the difference in structure between the synthesized polymer and the initial reactants. The shift and absence of some absorption bands from the characteristic functional groups of monomers was observed, which indicates the creation of a new structure. A broad absorption band of high intensity from valence vibrations of the N-H bond in the FTIR spectrum of copolymer at 3442 cm^−1^ is shifted to higher wavenumbers compared to the same band in the FTIR spectrum of monomer NiPMAm (Figure 8a) and comonomer NiPAm (Figure 8b). The displacement of the centroid of this band indicates the involvement of the amino group in the formation of the hydrogen bond. This fact is supported by displacement of amide band II compared to its placement in the FTIR spectrum of the monomer. An absorption band of the copolymer related to valence vibrations of the C=O group (amide band I) appears at 1649 cm^−1^ and shifted to lower wavenumbers compared to the absorption band in the FTIR spectra of NiPMAm and NiPAm by 4 and 10 units, respectively. The absorption bands at 1387 and 1367 cm^−1^ confirm the presence of an isopropyl group in the structure of the copolymer p(NiPMAm/NiPAm) and indicate that this group, that is present in monomers, was not involved in the polymerization process. The absence of an absorption band in the valence vibrations of the C=C bond, and which for monomers appears around wave numbers in the range of 1600–1640 cm^−1^, clearly indicates that the polymerization reaction of the NiPMAm and NiPAm monomers happened by the breaking of double bonds.

In the FTIR spectrum of curcumin (Figure 9a), a broad absorption band appears at 3421 cm^−1^ as a result of the valence vibrations of the free phenolic OH group. In the wavenumber range of 2800–3000 cm^−1^ in the FTIR spectrum of curcumin, two characteristic absorption bands appear with a maximum at 2967 and 2841 cm^−1^ that come from the asymmetric valence vibrations of the methyl group and the valence vibrations of the methoxy group, respectively [50,51,52]. The presence of an aromatic structure is confirmed by the absorption bands of valence vibrations of the C=C group which appears in the wavenumber range of 1600–1450 cm^−1^, and in the FTIR spectrum of curcumin are present with a maximum at 1627, 1590, 1512 and 1459 cm^−1^. In accordance with literature data, the absorption band with a maximum at 1627 cm^−1^ can be related to valence vibrations of the carbonyl group [53]. The absorption band with a maximum at 1209 cm^−1^ comes from valence vibrations of the C-O phenolic group, while the absorption band that comes from the C-O-C bond in the FTIR spectrum of curcumin appears at 1031 cm^−1^ [54]. In the wavenumber range 730–860 cm^−1^ absorption bands are present which come from deformative vibrations out of the C-H plane of the aromatic ring.

The broad absorption band, with a maximum of absorption at 3465 cm^−1^ in the FTIR spectrum of 2-hydroxypropyl-β-cyclodextrin (Figure 9b), is a result of valence vibrations of the hydroxyl group from 2-hydroxypropyl-β-cyclodextrin. The absorption bands coming from valence vibrations of C-H bound are appearing in the FTIR spectrum with a maximum at 2928 and 2969 cm^−1^ [55]. In the FTIR spectrum of 2-hydroxypropyl-β-cyclodextrin, the absorption band with a maximum at 1623 cm^−1^ comes from O-H deformative vibration in the plane [56]. The asymmetric and symmetric C–H deformative vibrations in the plane give absorption bands at 1458 and 1371 cm^−1^ in the FTIR spectrum of 2-hydroxypropyl-β-cyclodextrin, respectively. The absorption band at 1155 cm^−1^ indicates the valence vibrations of the C–C group, while absorption bands with maximums at 1083 and 1037 cm^−1^ confirm the presence of valence vibrations of the C–O bond at ether and hydroxyl groups of 2-hydroxypropyl-β-cyclodextrin. In the range of 1000–700 cm^−1^ valence and deformation vibration bands of glucopyranose units appear [57].

By comparative analysis of FTIR spectra of curcumin and 2-hydroxypropyl-β-cyclodextrin with the FTIR spectrum of curcumin: 2-hydroxypropyl-β-cyclodextrin complex (Figure 9c), changes in appearance of FTIR spectra of complex compared to the FTIR spectra of pure substances are noticed. The broad absorption band with the maximum of absorption at 3420 cm^−1^ in the FTIR spectrum of curcumin: 2-hydroxypropyl-β-cyclodextrin complex (Slika 9c) is shifted to lower wavenumbers by 45 units compared to the position of the same absorption band in the FTIR spectrum of 2-hydroxypropyl-β-cyclodextrin (Slika 9b). In the FTIR spectrum of complex of curcumin: 2-hydroxypropyl-β-cyclodextrin the absence of two characteristic absorption bands with maximums of absorptions at 2967 and 2841 cm^−1^ that come from asymmetric and symmetric vibrations of methyl group and valence vibrations of methoxy group of curcumin (Figure 9a), respectively, is noticed. The absorption bands that come from C-H valence vibrations with the maximums at 2928 and 2969 cm^−1^ in the FTIR spectrum of 2-hydroxypropyl-β-cyclodextrin are shifted by 3 units to lower i.e., by 2 units to higher wavelength numbers in the FTIR spectrum of curcumin: 2-hydroxypropyl-β-cyclodextrin (Figure 9c) and appear at 2925 cm^−1^ and 2971 cm^−1^, respectively. These changes may indicate the interaction of these groups from 2-hydroxypropyl-β-cyclodextrin with appropriate groups from the molecule of curcumin. The absorption band which comes from C-O-C bond in the FTIR spectrum of curcumin and appears at 1031 cm^−1^ is not present in the FTIR spectrum of curcumin: 2-hydroxypropyl-β-cyclodextrin complex (Figure 9c). The absorption band of C–O valence vibrations from 2-hydroxypropyl-β-cyclodextrin in the FTIR spectrum of complex is shifted by 2 units to higher wavelength numbers. The absence and shift of some absorption bands in the FTIR spectrum of curcumin: 2-hydroxypropyl-β-cyclodextrin complex indicate the incorporation of curcumin in holes of cyclodextrin, which is in accordance with literature data [58].

By incorporation of curcumin into p(NiPMAm/NiPAm) hydrogels, it is expected that the establishment of intermolecular interactions of the type hydrogen bondage between phenolic OH groups of curcumin as a proton-donor, with oxygen from the C=O group as a proton-acceptor of side chains of p(NiPMAm/NiPAm) copolymer occur. In addition, the C=O group of curcumin can form hydrogen bonds with NH proton donor groups of side chains of p(NiPMAm/NiPAm) hydrogels.

In the FTIR spectrum of p(NiPMAm/NiPAm), hydrogels containing 2 mol% of crosslinker with incorporated curcumin: 2-hydroxypropyl-β-cyclodextrin inclusion complex (Figure 10b), at the wavelength number range of 3500–3200 cm^−1^ broad absorption band with the maximum at 3430 cm^−1^ that comes from valence vibrations of N-H bond of hydrogels and valence vibrations of the phenolic OH group of curcumin, can be noticed. The maximum of this band is shifted by 12 units to lower wavelength numbers compared to the position of the same absorption band in the FTIR spectrum of empty hydrogel (Figure 10a), and by 9 units to higher wavelength numbers compared to the position of the absorption band in the FTIR spectrum of curcumin (Figure 9a). This indicates that the mentioned groups are involved in the formation of hydrogen bonds between curcumin molecules and the hydrogel. The amide absorption band I, ν(C=O), appears at 1653 cm^−1^, while the amide absorption band II, δ(N-H), appears at 1549 cm^−1^, and their maximums are shifted by 4, that is, 6 units to higher wavelength numbers, compared to the spectrum of hydrogel only (Figure 10a). The maximum of absorption that comes from valence vibrations of C=O in the FTIR spectrum of hydrogels with incorporated curcumin (Figure 10b) is shifted by 26 units to higher wavelength numbers, compared to the position of the same absorption band in the FTIR spectrum of curcumin (Figure 9a). The shiftment of the maximum of amide absorption bands and the decrease in their intensity compared to hydrogel only, indicates the involvement of C=O and –NH groups in the formation of hydrogen bonds. The absorption band that comes from valence vibrations of the C-O-C bond is present with a maximum at 1034 cm^−1^ and is shifted to higher wavelength numbers by 3 units, compared to the position of the same band in the FTIR spectrum of curcumin. The results of FTIR analysis show that the change in intensity and the shift of the characteristic absorption bands of curcumin and hydrogel to lower or higher values of wavelength numbers happened, which indicates the incorporation of curcumin into the hydrogel structure.

Shifts of the corresponding absorption maxima by several units of cm^−1^ indicate the formation of weak hydrogen bonds, which is favorable from the aspect of formulation development. This indicates that these weak hydrogen bonds will possibly slow down the diffusion of curcumin molecules from the hydrogel structure, and enable the release of curcumin from the formulation for a longer period of time.

### 3.5. Scanning Electron Microscopy (SEM)

The morphology of curcumin, curcumin: 2-hydroxypropyl-β-cyclodextrin complex, synthesized p(NiPMAm/NiPAm) hydrogel containing 2 mol% crosslinker and p(NiPMAm/NiPAm) hydrogel with incorporated complex of curcumin: 2-hydroxypropyl-β-cyclodextrin, was examined by SEM method. Hydrogel samples were swollen to equilibrium and then lyophilized in order to better understand their morphology. Obtained SEM micrographs are shown in Figure 11.

In Figure 11a curcumin crystals are clearly visible, while in Figure 11b crystal structure of curcumin: 2-hydroxypropyl-β-cyclodextrin complex cannot be seen. By analyzing the SEM micrographs, a clear difference between the look of the surface structure of the empty hydrogel (Figure 11c) and hydrogel with incorporated curcumin: 2-hydroxypropyl-β-cyclodextrin complex can be observed (Figure 11d). The structure of p(NiPMAm/NiPAm) hydrogel containing 2 mol% EGDM is porous, with a pore diameter in the range of 100 to 300 μm (Figure 11c). In micrography 7d, the presence of curcumin: 2-hydroxypropyl-β-cyclodextrin complex in holes of hydrogel can be seen, whose structure matches the structure of curcumin: 2-hydroxypropyl-β-cyclodextrin complex (Figure 11b). This indicates a successful incorporation of curcumin: 2-hydroxypropyl-β-cyclodextrin inclusion complex into holes of the synthesized p(NiPMAm/NiPAm) gel, and is in accordance with the result obtained by FTIR analysis.

### 3.6. Differential Scanning Calorimetry (DSC)

In Figure 12 are shown DSC curves of curcumin, 2-hydroxypropyl-β-cyclodextrin, curcumin: 2-hydroxypropyl-β-cyclodextrin complex, p(NiPMAm/NiPAm) 10/90/2 hydrogel and p(NiPMAm/NiPAm) 10/90/2 hydrogel with incorporated curcumin: 2-hydroxypropyl-β-cyclodextrin complex. In the DSC curve that comes from curcumin (curve 1 in Figure 12) can be seen the endothermic melting peak of curcumin at 188 °C and the peak area correspond to the curcumin melting enthalpy of 171.5 J/g. Considering that curcumin complexation occurred in the presence of 2-hydroxypropyl-β-cyclodextrin, the curcumin molecules in the complex and in the gel complex formulation cannot form curcumin crystals that would give an endothermic melting peak. Nevertheless, in the endothermic peak of the complex, a weak peak is observed at 186 °C, which may indicate that the curcumin molecules have not completely entered the holes of 2-hydroxypropyl-β-cyclodextrin (curve 3 in Figure 12). In the formulation of the complex with gel (curve 2 in Figure 12), the endothermic peak from the melting of curcumin can no longer be observed. This indicates that the supramolecular structures of the complexes are now already distributed in the gel, and that a constant release rate of curcumin could be expected from this formulation of the curcumin complex in the gel.

### 3.7. X-ray Diffraction (XRD)

The XRD spectra of curcumin, 2-hydroxypropyl-β-cyclodextrin, curcumin: 2-hydroxypropyl-β-cyclodextrin complex, synthesized p(NiPMAm/NiPAm) hydrogel containing 2 mol% EGDM and p(NiPMAm/NiPAm) hydrogel with incorporated curcumin: 2-hydroxypropyl-β-cyclodextrin complex are shown in Figure 13.

In the X-ray diffractogram of curcumin (Figure 13a), sharp peaks are present at the values of diffraction angle 2 θ: 7.6; 8.7; 16; 17.5; 21 and 42.55° that indicates that pure curcumin is in crystalline form, which is in agreement with the literature data [22]. Two broad peaks without high maximums that appear in the range of 2 θ = 5–15° and 2 θ = 15–22.5° in diffractogram of 2-hydroxypropyl-β-cyclodextrin (Figure 13b) indicate its amorphous structure [59]. By comparative analysis of diffractograms in Figure 13, it can be observed that in the diffractogram of the inclusion complex of curcumin: 2-hydroxypropyl-β-cyclodextrin (Figure 13c), there are no sharp peaks present in the diffractogram of pure crystalline curcumin. The appearance of new peaks in the diffractogram of the inclusion complex clearly indicates the formation of a new supramolecular structure. Sharp peaks at the values of the diffraction angle 2 θ: 7, 13.5, 16, 19 and 27° in diffractogram of the curcumin: 2-hydroxypropyl-β-cyclodextrin complex indicate that the curcumin molecule did not completely enter into the hole of the 2-hydroxypropyl-β-cyclodextrin from inclusion complex. This is understandable, since one molecule of curcumin requires two molecules of 2-hydroxypropyl-β-cyclodextrin to be completely included in the cavities of 2-hydroxypropyl-β-cyclodextrin [24]. However, that was not necessary in this case because the further homogenization was carried over by the gel. By comparative analysis of the diffractogram of the empty hydrogel and the hydrogel with incorporated complex of curcumin: 2-hydroxypropyl-β-cyclodextrin (Figure 13d and e, respectively), great similarity was observed considering the shape and position of the peaks that appear as broad in the range of 2 θ = 5–10 and 2 θ = 15–32.5, and they correspond to the peaks that appear in the diffractogram of the empty hydrogel. This indicates that the structure of the hydrogel after incorporation of the inclusion complex did not significantly change, and that the complex of curcumin: 2-hydroxypropyl-β-cyclodextrin was relatively uniformly distributed into the gel.

### 3.8. Nuclear Magnetic Resonance (^1^H-NMR)

In Figure 14 are shown ^1^H-NMR spectra of 2-hydroxypropyl-β-cyclodextrin and the complex of curcumin: 2-hydroxypropyl-β-cyclodextrin. In Table 3 are shown values for chemical shifts of 2-hydroxypropyl-β-cyclodextrin and the complex of curcumin: 2-hydroxypropyl-β-cyclodextrin, and the change in chemical shifts of protons which originate from 2-hydroxypropyl-β-cyclodextrin in the complex. Considering that the solubility of curcumin in water is very weak, the analysis is aimed at monitoring the chemical shifts of protons from 2-hydroxypropyl-β-cyclodextrin. The changes in chemical shifts of protons Δδ which are shown in Table 3, indicate the formation of weak hydrogen bonds in which protons from 2-hydroxypropyl-β-cyclodextrin are involved which indicates the possible inclusion and creation of the inclusion complex.

### 3.9. The Loading Efficiency of Curcumin into the p(NiPMAm/NiPAm) Hydrogel

By knowing that curcumin does not dissolve in aqueous media, the incorporation of curcumin into the hydrogel was accomplished by incorporating the curcumin: 2-hydroxypropyl-β-cyclodextrin complex into samples of hydrogels. The efficiency of curcumin incorporation, η, was calculated according to Equation (2), in terms of the total starting mass of curcumin available in the complex. The data are shown in Table 4.

The efficiency of curcumin incorporation into the inclusion complex of p(NiPMAm/NiPAm) 10/90/2 xerogel is greater than for 10/90/3, that is in agreement with the results for the swelling for synthesized gels. These values are satisfying considering that the water solubility of curcumin is very low. By complexation with 2-hydroxypropyl-β-cyclodextrin, its water solubility is increased 1237 times, and this provides easier incorporation into synthesized hydrogels as well as a release in physiological mediums.

### 3.10. In Vitro Release of Curcumin from p(NiPMAm/NiPAm) Gels

The release of curcumin from p(NiPMAm/NiPAm) gels containing 2 and 3 mol% of EGDM was monitored under in vitro conditions at temperature 37 °C and pH 7.4, that simulate the body temperature and pH conditions as in the small intestine, by applying the HPLC method (Figure 15). The release of curcumin from the gels was monitored during 48 h.

The release rate of curcumin from p(NiPMAm/NiPAm) hydrogels containing 2 mol% of EGDM is 13.13 µg/h, and for hydrogels containing 3 mol% of EGDM it is 2.51 µg/h, which provides a prolonged release of curcumin during 366.5 h (15.27 days) and 1653.6 h (68.9 days), respectively. In Figure 15a it can be seen that there is also an initial release of curcumin in an amount of approximately 1600 µg from the formulation of sample 10/90/2, and approximately 500 µg from the formulation of sample 10/90/3 (Figure 15b). The results obtained indicate the possibility that thermosensitive p(NiPMAm/NiPAm) hydrogels could find application in the development of formulations with the prolonged release of curcumin. These results shows that there is a possibility of tailoring the formulation in terms of the amount of curcumin that should be incorporated into the gel through the complex, for tailoring the time and rate of curcumin release from this formulation as well as the intensity of the initial release; for example, by changing the crosslinker concentration. The amount of crosslinker for synthesis of gel will determine the density of the nodes in the network and the length of the branches in the gel network, that will affect the diffusion of curcumin molecules through the mass of gel. In a polymer network created with a higher concentration of crosslinker, the branches will be shorter and the concentration of nodes will be higher, and like this the diffusion of curcumin molecules will slow down. Analysis of the release mechanism of curcumin from the formulation of the complex in gel can be helpful for this purpose.

With the aim to further study the curcumin release mechanism from p(NiPMAm/NiPAm), hydrogels with the incorporated inclusion complex of curcumin: 2-hydroxypropyl-β-cyclodextrin, the experimental data obtained in this work are fitted by appropriate mathematical models: Higuchi, Korsmeyer–Peppes and Baker–Lonsdale, and the obtained parameters are shown in Table 5.

The highest coefficient of determination (R^2^) and the lowest AIC indicate that Korsmeyer-Peppes’s model best describes the release of curcumin from the formulation of p(NiPMAm/NiPAm) hydrogels containing curcumin: 2-hydroxypropyl-β-cyclodextrin complex. The value for the diffusion exponent of the hydrogel sample p(NiPMAm/NiPAm) containing 2 mol% EGDM is 0.074, but 0.063 for the sample of p(NiPMAm/NiPAm) hydrogel containing 3 mol% of EGDM, and that shows that the release mechanism of curcumin, is based on diffusion from the polymer matrix of the gels. This definitely confirms the conclusion that the right choice on the crosslinker concentration to be used for the synthesis process of p(NiPMAm/NiPAm) hydrogels can have an influence on the release rate of curcumin from the gel formulation with curcumin: 2-hydroxypropyl-β-cyclodextrin complex. The form of curcumin: 2-hydroxypropyl-β-cyclodextrin complex was applied to provide a larger amount of curcumin into the gel.

If the overall results are considered, it can be concluded that a system for the sustained release of curcumin from a formulation base on hydrogel has been developed. This was achieved primarily by increasing solubility by including curcumin to 2-hydroxypropyl-β-cyclodextrin, which enabled the required amount of curcumin to enter into the hydrogel. Incorporation of pure curcumin into the hydrogel would create curcumin agglomerates in the formulation, while the cyclodextrin complex enabled individual curcumin molecules to diffuse through the gel. By that, the entire formulation significantly contributed to increasing the solubility of curcumin.

## 4. Conclusions

In this work, the synthesis and characterization of p(NiPMAm/NiPAm) hydrogels with two different concentrations of crosslinker was done. The results show that less than 0.3% of residual monomers is present in synthesized hydrogels. The FTIR analysis of reactants at the beginning of the synthesis, synthesized hydrogels, curcumin: 2-hydroxypropyl-β-cyclodextrin complex and the formulation of hydrogel with complex of curcumin: 2-hydroxypropyl-β-cyclodextrin showed that the complexation and formulation of complexes with the gel were done by the help of hydrogen bonds formation. The water swelling process is controlled by the mechanisms called “non-Fick’s diffusion“ and “less to Fick’s diffusion“. The phase solubility of curcumin in solution of 2-hydroxypropyl-β-cyclodextrin showed increased solubility by 1237 times. The SEM analysis showed the loss of the crystal structure of curcumin in the complex and in the formulation of the complex with gel, which was additionally confirmed by DSC and XRD analyses. Since the water solubility of curcumin is low (3 mg/dm^3^), the complexation with 2-hydroxypropyl-β-cyclodextrinthe made incorporation of curcumin into hydrogels easier by increasing the solubility. The monitoring of the curcumin release profile from the formulation of curcumin: 2-hydroxypropyl-β-cyclodextrin complex with p(NiPMAm/NiPAm) hydrogels, shows the starting release of curcumin from the formulation that slows with the increasement of crosslinker in the composition of the reaction mixture for hydrogel synthesis. In addition, with an increase in the concentration of the crosslinker, the release rate of curcumin from the formulation also decreases, which gives the possibility of tailoring the release rate from the formulation. The release rate of curcumin from the formulation of the complex with gel is constant in the function of time, and is dependent on the amount of curcumin in the formulation and the release rate. This is a formulation from which the curcumin will be released over a longer period of time can be designed, more than over 60 days. The kinetic analysis of data for curcumin release from the formulation of curcumin-gel complex showed that the mechanism of curcumin release is based on diffusion from the polymer matrix of the gels.

## 5. Patents

Patent Application RS2022P0287, Urošević, M.; Nikolić, Lj.; Gajić, I.; Nikolić, V.; Dinić, A.; Ilić-Stojanović, S.; Miljković, V.; Nikolić, G.; Cakić, S. Formulation of the matrix system with curcumine, Priority 17 Mart 2022, the Intellectual Property Office of the Republic of Serbia.

## Figures and Tables

**Figure 1 pharmaceutics-15-00382-f001:**
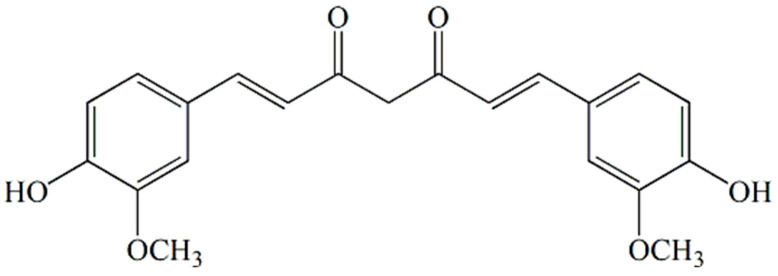
Chemical structure of curcumin.

**Figure 2 pharmaceutics-15-00382-f002:**
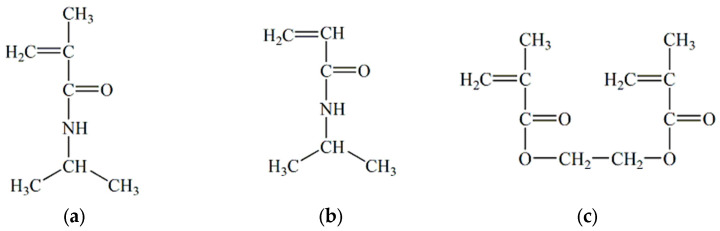
Chemical structure: (**a**) N-isopropylmethacrylamide, (**b**) N-isopropylacrylamide and (**c**) ethylene glycol dimethacrylate.

**Figure 3 pharmaceutics-15-00382-f003:**
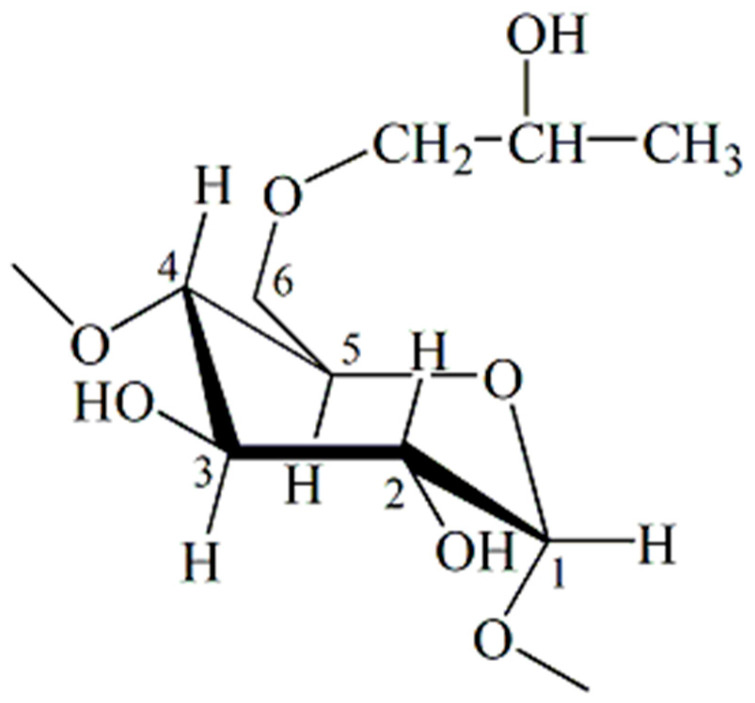
Chemical structure of a seven-membered ring molecule part of 2-hydroxypropyl-β-cyclodextrin.

**Figure 4 pharmaceutics-15-00382-f004:**
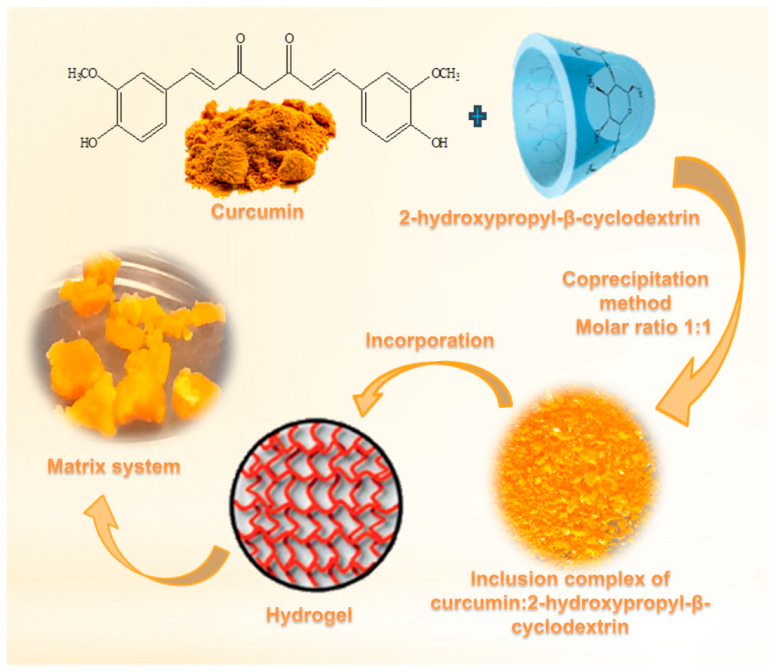
The schematic view of obtaining the formulation of curcumin: 2-hydroxypropyl-β-cyclodextrin complex with p(NiPMAm/NiPAm) hydrogel.

**Figure 5 pharmaceutics-15-00382-f005:**
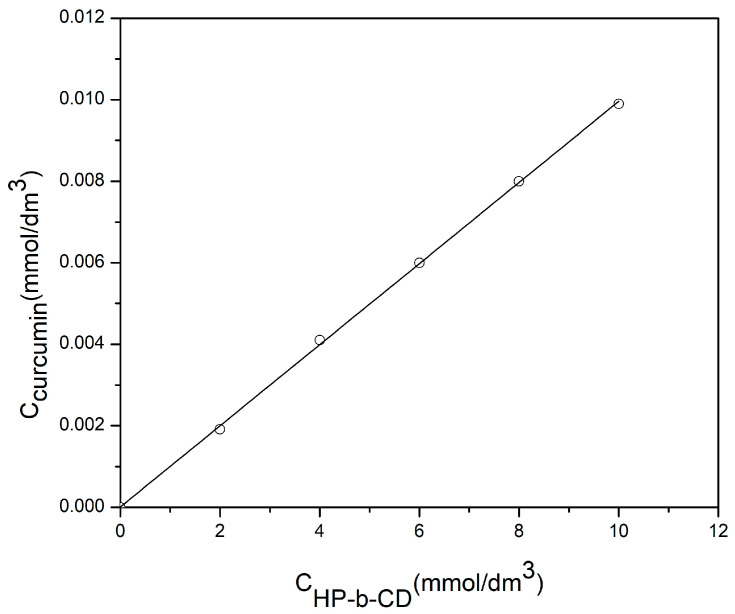
Phase solubility diagram of curcumin in the presence of 2-hydroxypropyl-β-cyclodextrin in water solution at 25 °C.

**Figure 6 pharmaceutics-15-00382-f006:**
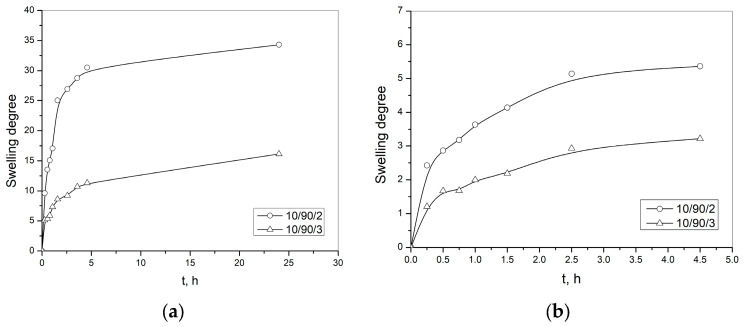
Dependence of the degree of swelling on time of p(NiPMAm/NiPAm) hydrogels in the solution which pH value is 7.4 at the temperature: (**a**) 25 °C and (**b**) 37 °C.

**Figure 7 pharmaceutics-15-00382-f007:**
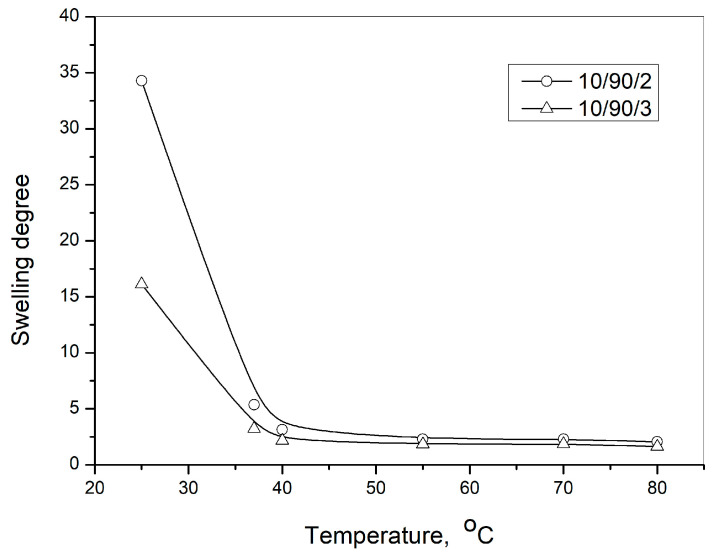
Dependence of the degree of swelling on temperature in the solution in which pH value is 7.4.

**Figure 8 pharmaceutics-15-00382-f008:**
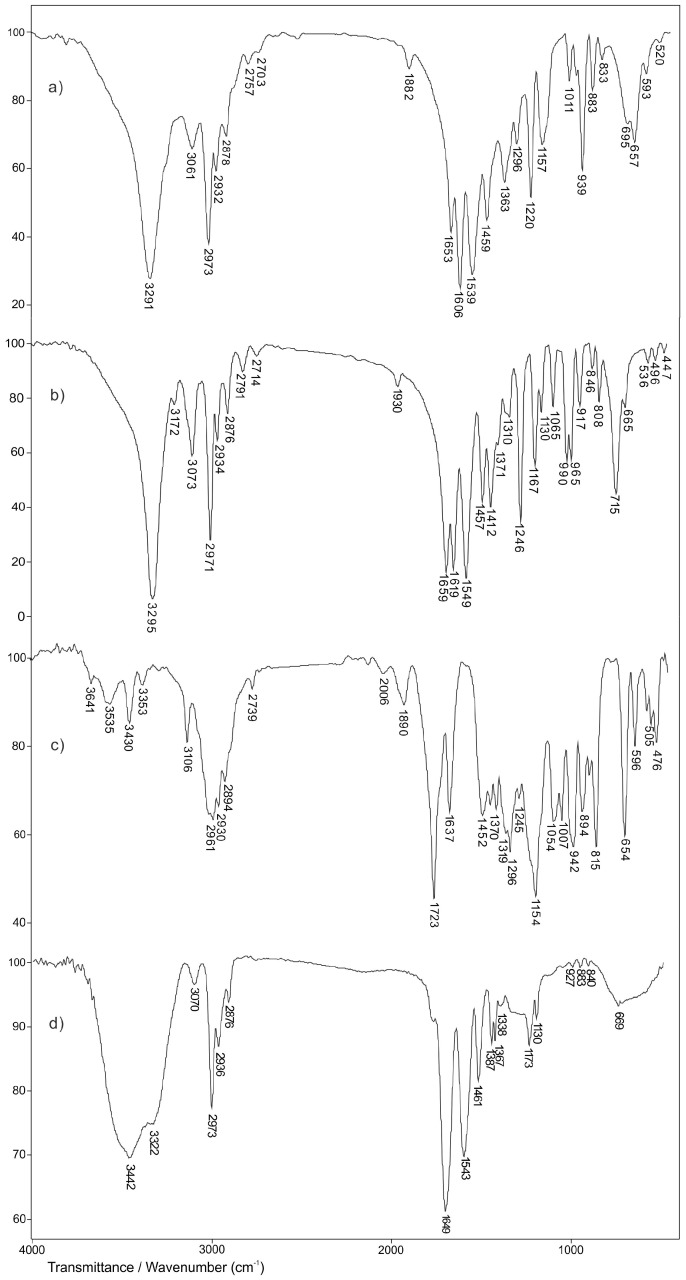
FTIR spectra: (**a**) NiPMAm monomer, (**b**) NiPAm monomer, (**c**) crosslinker EGDM and (**d**) synthesized p(NiPMAm/NiPAm) copolymer with 2 mol% of EGDM.

**Figure 9 pharmaceutics-15-00382-f009:**
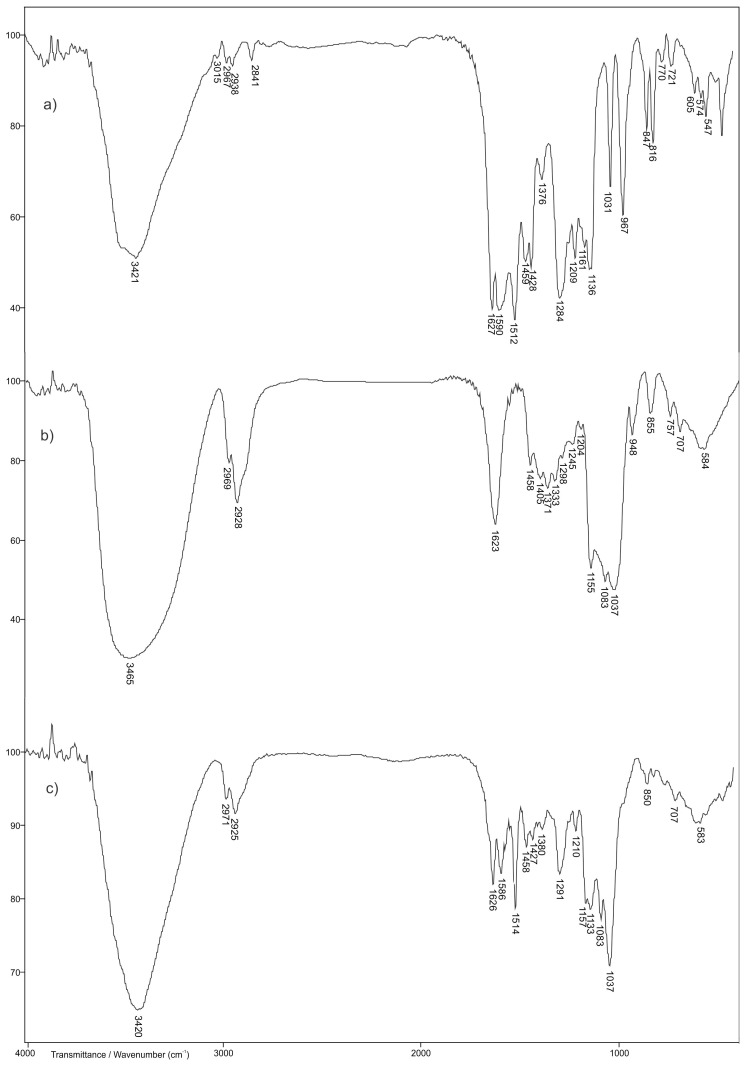
The FTIR spectra: (**a**) curcumin, (**b**) 2-hydroxypropyl-β-cyclodextrin, (**c**) curcumin: 2-hydroxypropyl-β-cyclodextrin inclusion complex.

**Figure 10 pharmaceutics-15-00382-f010:**
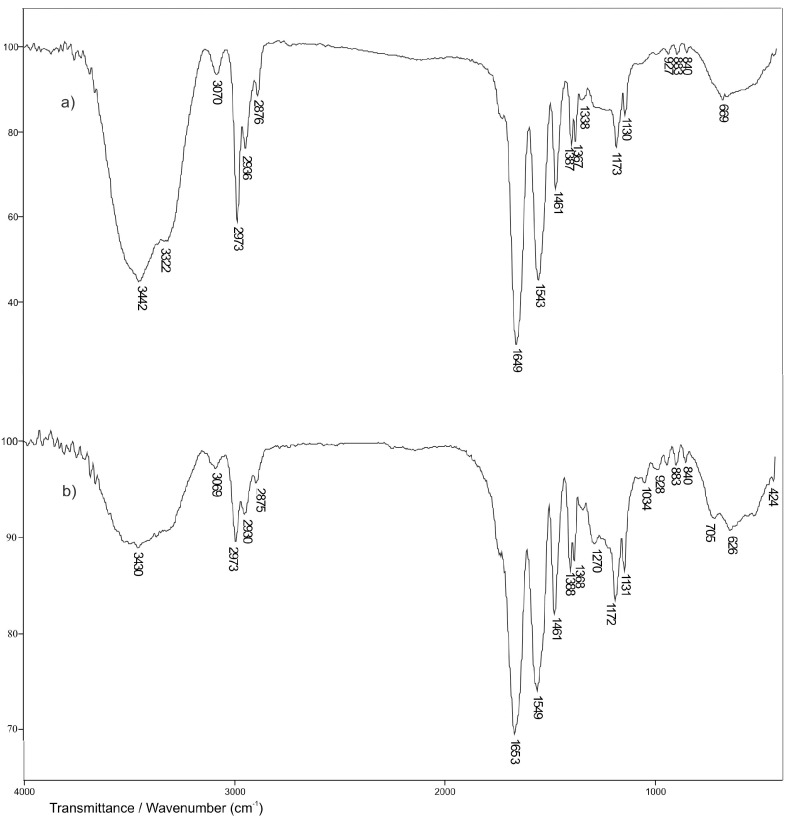
The FTIR spectra: (**a**) copolymer p(NiPMAm/NiPAm), sample 10/90/2 and (**b**) matrix system of p(NiPMAm/NiPAm) with incorporated inclusion complex of curcumin: 2-hydroxypropyl-β-cyclodextrin.

**Figure 11 pharmaceutics-15-00382-f011:**
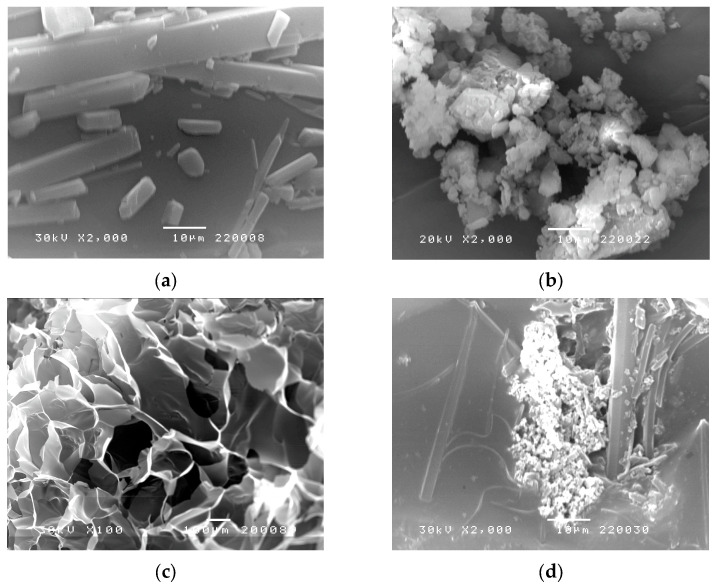
Scanning electron microscopy micrographs (SEM): (**a**) curcumin, (**b**) curcumin: 2-hydroxypropyl-β-cyclodextrin complex, (**c**) p(NiPMAm/NiPAm) hydrogel containing 2 mol% EGDM, (**d**) p(NiPMAm/NiPAm) hydrogel containing 2 mol% of crosslinker with incorporated curcumin: 2-hydroxypropyl-β-cyclodextrin complex.

**Figure 12 pharmaceutics-15-00382-f012:**
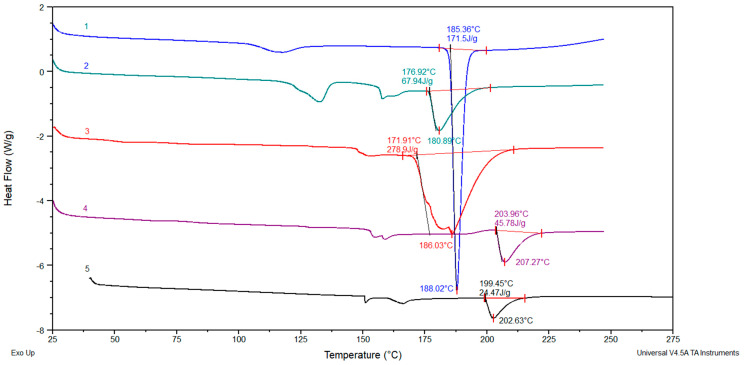
DSC curves: 1. curcumin, 2. p(NiPMAm/NiPAm) 10/90/2 hydrogel containing curcumin: 2-hydroxypropyl-β-cyclodextrin complex, 3. curcumin: 2-hydroxypropyl-β-cyclodextrin complex, 4. p(NiPMAm/NiPAm)10/90/2 hydrogel, 5. 2-hydroxypropyl-β-cyclodextrin.

**Figure 13 pharmaceutics-15-00382-f013:**
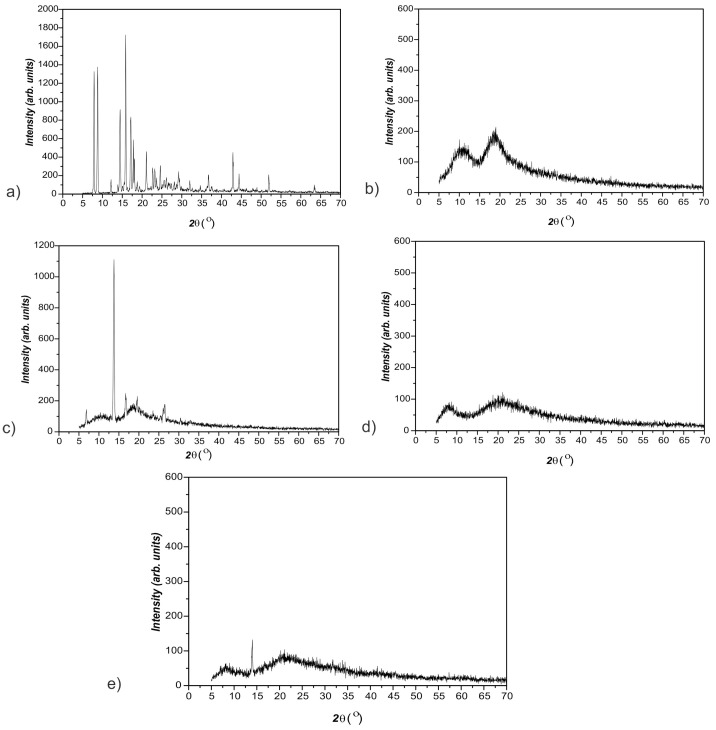
The XRD diffractograms: (**a**) curcumin, (**b**) 2-hydroxypropyl-β-cyclodextrin, (**c**) inclusion complex of curcumin: 2-hydroxypropyl-β-cyclodextrin, (**d**) p(NiPMAm/NiPAm) hydrogel containing 2 mol% of crosslinker, and (**e**) p(NiPMAm/NiPAm) hydrogel with incorporated curcumin: 2-hydroxypropyl-β-cyclodextrin complex.

**Figure 14 pharmaceutics-15-00382-f014:**
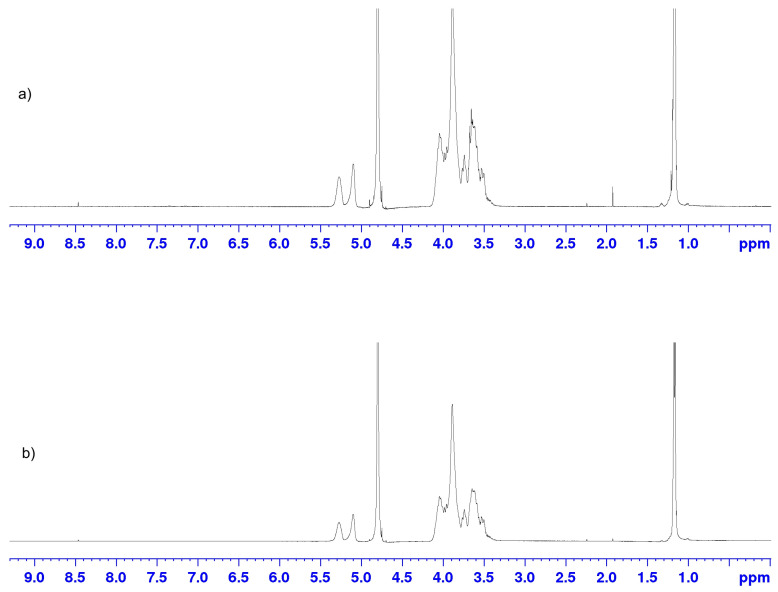
^1^H-NMR spectrum: (**a**) complex of curcumin: 2-hydroxypropyl-β-cyclodextrin and (**b**) 2-hydroxypropyl-β-cyclodextrin.

**Figure 15 pharmaceutics-15-00382-f015:**
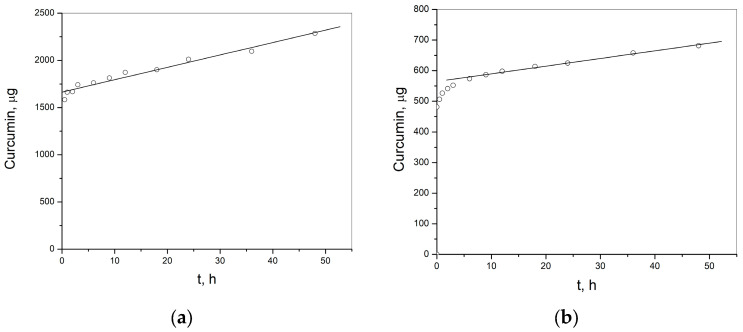
Profile of curcumin release from p(NiPMAm/NiPAm) hydrogels: (**a**) sample 10/90/2 and (**b**) sample 10/90/3.

**Table 1 pharmaceutics-15-00382-t001:** Kinetic parameters of swelling of p(NiPMAm/NiPAm) hydrogels at pH 7.4.

Temperature, °C	Sample	*n*	k, min^1/*n*^	D, cm^2^/min
25	10/90/2	0.82	0.092	1.65 ∙ 10^−5^
	10/90/3	0.60	0.127	3.17 ∙ 10^−5^
37	10/90/2	0.49	0.231	1.05 ∙ 10^−4^
	10/90/3	0.63	0.164	5.26 ∙ 10^−5^

*n*-diffusion exponent; *k*-the constant characteristic for certain types of polymer network (min^1/*n*^); *D*-diffusion coefficient (cm^2^/min).

**Table 2 pharmaceutics-15-00382-t002:** The content of residual monomers in synthesized p(NiPMAm/NiPAm) hydrogels.

Sample	The Content of Residual Monomers in Sample, %		
	NiPMAm	NiPAm	EGDM
10/90/2	0.11	0.28	0.07
10/90/3	0.13	0.30	0.08

**Table 3 pharmaceutics-15-00382-t003:** Chemical shifts values δ for 2-hydroxypropyl-β-cyclodextrin (2-HP-β-CD) and for complex of curcumin: 2-hydroxypropyl-β-cyclodextrin (Complex) and changes in chemical shifts Δδ for protons originate from 2-hydroxypropyl-β-cyclodextrin in complex.

Type of Proton	Chemical Shifts Values δ		Δδ
	2-HP-β-CD	Complex	
CH_3_	1.1692	1.1678	+0.0014
H-C_1_	5.1005	5.0993	+0.0012
H-C_2_	5.2732	5.2743	−0.0011

**Table 4 pharmaceutics-15-00382-t004:** The efficiency of curcumin incorporation (η) into p(NiPMAm/NiPAm) xerogels.

Sample	η of Curcumin (%)
10/90/2	78.35
10/90/3	67.58

**Table 5 pharmaceutics-15-00382-t005:** The kinetic of curcumin release from p(NiPMAm/NiPAm) hydrogels.

Kinetic Model	Parameter	Sample 10/90/2	Sample 10/90/3
Higuchi 𝐹 = 𝑘_𝐻_∙𝑡^1/2^	*k_H_*	9.218	3.350
	R^2^	−0.918	−1.147
	AIC	97.98	74.811
Korsmeyer–Peppas 𝐹 = 𝑘_𝐾𝑃_∙𝑡^𝑛^	*k_KP_*	33.247	12.499
	*n*	0.074	0.063
	R^2^	0.984	0.996
	AIC	41.036	−2.927
Baker–Lonsdale 3/2[1−(1−𝐹/100)^2/3^]−𝐹/100 =𝑘 _𝐵𝐿_∙𝑡	*k_BL_*	0.002	0
	R^2^	−0.636	−1.054
	AIC	96.078	74.27

*k*_H_: the constant of release at Higuchi model; R^2^: coefficient of determination; AIC: Akaik’s information criterion; F: the fraction of released drug in function of time *t*; *k*_KP_: the constant of release at Korsmeyer-Peppes model that takes into account structural and geometrical characteristics of dosage form; *n*: the exponent of release/diffusion; *k*_BL_: combined constant of release at Baker-Lonsdale model; [60,61,62].

## Data Availability

Not applicable.

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
