# Peer review of "The Formulation of Curcumin: 2-Hydroxypropyl-β-cyclodextrin Complex with Smart Hydrogel for Prolonged Release of Curcumin"

_pharmaceutics, 2023, doi:10.3390/pharmaceutics15020382_

Round 1
Reviewer 1 Report
The manuscript entitled “The formulation of curcumin:2-hydroxypropyl-β-cyclodextrin complex with smart hydrogel for prolonged release of curcumin”, authored Nikolić et al., deals with preparation of new matrix system based on polymer carrier crosslinked with poly(N-isopropylmethacrylamide-co-N- isopropy-lacrylamide) with an incorporated inclusion complex of curcumin:2- hydroxypro-pyl-β-cyclodextrin to increase solubility and bioavailability of curcumin. The topic of this manuscript is important and current, and results could be interesting for readers. However, some changes have to be entered into the revised version of the manuscript before it can be further processed:
1. Authors create curcumin:2-hydroxypropyl-β-cyclodextrin inclusion complex. The reference method for creating the inclusion complex is NMR, which should be provided.
2. The authors create a new combination of p(NiPMAm/NiPAm) hydrogel with curcumin:2-hydroxypropyl-β-cyclodextrin complex - the toxicity of system should be checked.
3. As the aim of the work, the authors want to increase the bioavailability of curcumin. However, there are no tests at work confirming this. Please provide such tests. If no, please indicate only the improvement of solubility as the objective.
Author Response
- Authors create curcumin:2-hydroxypropyl-β-cyclodextrin inclusion complex. The reference method for creating the inclusion complex is NMR, which should be provided.
We are grateful for the reviewer's suggestion. NMR spectra were recorded and showed in the manuscript, and changes in chemical shift of protons originate from 2-hydroxypropyl-β-cyclodextrin (Table 3) showed that they contribute to creation of hydrogen bond with guest molecule, which is a confirmation for inclusion complex creation.
Also, X-ray diffraction shows change in XRD spectra, Figure 12c - inclusion complexes, which differs from Figure 12a - curcumin. Namely, with the inclusion, many reflection planes of pure curcumin are lost, which is also a confirmation of the inclusion.
- The authors create a new combination of p(NiPMAm/NiPAm) hydrogel with curcumin:2-hydroxypropyl-β-cyclodextrin complex - the toxicity of system should be checked.
Considering that curcumin is ubiquitous in the human diet, that many inclusion complexes with cyclodextrins and their derivatives for pharmaceutical use were made, that mentioned hydrogel based on N-isopropylmethacrylamide or N-isopropylacrylamide have also been used for pharmaceutical formulations, that none of this components didnt showed toxicity in low amounts and that no new compounds are formed by their mutual reaction, rather only weak hydrogen type bonds have been formed, we considered that toxicity testing was not necessary at this time. Also, I have to admit that we dont have experimental conditions to do that quickly, so I kindly ask the reviewers and editors not to insist on this test. Since it is an extensive work to analyze the cytotoxicity of this formulation on several different cell lines of healthy and cancer cells, that will be the subject of future work and we can do that as a continuation of research.
- As the aim of the work, the authors want to increase the bioavailability of curcumin. However, there are no tests at work confirming this. Please provide such tests. If no, please indicate only the improvement of solubility as the objective.
We agree that the aim of the overall work is to increase the solubility of curcumin, and indirectly this means that with increasing solubility there is a possibility of increasing bioavailability if the curcumin from this formulation is released slowly and long enough. We agree that the aim of the work is to increase solubility and this is emphasized in the manuscript.
Sincerely,
Authors
Reviewer 2 Report
The novelty character of paper should be better marked as well as the advantages of the use of FTIR approach.
In Introduction properties and advances researches on curcumin should be better marked and related references added such as:
Zielińska et al. Medicina (Kaunas). 2020;56(7):336. doi:10.3390/medicina56070336
A graphical scheme of study approach should be inserted in Materials and Methods.
Results in Figure 5 and 6 should be better described in the text.
Results on Residual Monomers Analysis should be better discussed.
Author Response
The novelty character of paper should be better marked as well as the advantages of the use of FTIR approach.
We thank the reviewer for this suggestion. In addition to the formation of the inclusion complex of curcumin with 2-hydroxypropyl-β-cyclodextrin, which significantly increased the solubility of curcumin, that complex enabled enter of curcumin into the hydrogel. Curcumin can be released from the hydrogel at a constant rate for a long time, and this is the biggest contribution of these studies. The advantage of using the FTIR method for analysis is that it is, nowdays, a very simple and almost routine method of analysis, and it provides valuable data regarding the appropriate identification or allows conclusions to be drawn about the formation of weak hydrogen bonds between certain functional groups of substances if certain vibrations shift in the spectrum by several units cm-1. These corrections were made in the manuscript.
In Introduction properties and advances researches on curcumin should be better marked and related references added such as:
Zielińska et al. Medicina (Kaunas). 2020;56(7):336. doi:10.3390/medicina56070336
In the section introductory, a paragraph related to advanced curcumin research has been added, adequate references are listed, as well as a suggested reference.
A graphical scheme of study approach should be inserted in Materials and Methods.
A graphic scheme was included in the manuscript.
Results in Figure 5 and 6 should be better described in the text.
The discussion of the results in Figures 5 and 6 has been improved and added to the manuscript.
Results on Residual Monomers Analysis should be better discussed.
The analysis of residual monomers is improved. What the authors could not mention in the manuscript is that some of the co-authors have been dealing with the synthesis and characterization of such hydrogels for more than 10 years and that this experience contributed to the development of a synthesis procedure with variable temperatures that enables a low level of residual monomers in the hydrogel and thus and easier extraction from the hydrogel, if it is needed to obtain a hydrogel without residual monomers, as it is the case for pharmaceutical applications.
Sincerely,
Authors
Round 2
Reviewer 1 Report
Please remove the information about increasing bioavailability from the content of the manuscript. In the previous answer, the authors agreed with the note that they only study the effect on the solubility of the compound.
Author Response
Dear Editor, Reviewer,
we made corrections of the manuscript according to the suggestions of the reviewers.
Yours sincerely,
Ljubiša Nikolić